# Teeth Microcracks Research: Towards Multi-Modal Imaging

**DOI:** 10.3390/bioengineering10121354

**Published:** 2023-11-25

**Authors:** Irma Dumbryte, Donatas Narbutis, Maria Androulidaki, Arturas Vailionis, Saulius Juodkazis, Mangirdas Malinauskas

**Affiliations:** 1Institute of Odontology, Vilnius University, LT-08217 Vilnius, Lithuania; 2Institute of Theoretical Physics and Astronomy, Vilnius University, LT-10222 Vilnius, Lithuania; 3Microelectronics Research Group, Institute of Electronic Structure & Laser, Foundation for Research and Technology FORTH-Hellas, 70013 Heraklion, Crete, Greece; 4Stanford Nano Shared Facilities, Stanford University, Stanford, CA 94305, USA; 5Department of Physics, Kaunas University of Technology, LT-51368 Kaunas, Lithuania; 6Optical Sciences Centre and ARC Training Centre in Surface Engineering for Advanced Materials (SEAM), School of Science, Swinburne University of Technology, Hawthorn, VIC 3122, Australia; 7WRH Program International Research Frontiers Initiative (IRFI), Tokyo Institute of Technology, Nagatsuta-cho, Midori-ku, Yokohama 226-8503, Japan; 8Laser Research Center, Vilnius University, LT-10223 Vilnius, Lithuania

**Keywords:** artificial intelligence, biosolids, clinical diagnostics, enamel damage, machine learning, spectroscopy, X-ray micro-computed tomography

## Abstract

This perspective is an overview of the recent advances in teeth microcrack (MC) research, where there is a clear tendency towards a shift from two-dimensional (2D) to three-dimensional (3D) examination techniques, enhanced with artificial intelligence models for data processing and image acquisition. X-ray micro-computed tomography combined with machine learning allows 3D characterization of all spatially resolved cracks, despite the locations within the tooth in which they begin and extend, and the arrangement of MCs and their structural properties. With photoluminescence and micro-/nano-Raman spectroscopy, optical properties and chemical and elemental composition of the material can be evaluated, thus helping to assess the structural integrity of the tooth at the MC site. Approaching tooth samples having cracks from different perspectives and using complementary laboratory techniques, there is a natural progression from 3D to multi-modal imaging, where the volumetric (passive: dimensions) information of the tooth sample can be supplemented by dynamic (active: composition, interaction) image data. Revelation of tooth cracks clearly shows the need to re-assess the role of these MCs and their effect on the structural integrity and longevity of the tooth. This provides insight into the nature of cracks in natural hard materials and contributes to a better understanding of how bio-inspired structures could be designed to foresee crack propagation in biosolids.

The definition of enamel microcracks (MCs), introduced more than 20 years ago, referred to incomplete tooth fractures that usually do not cross the dentin–enamel junction (DEJ) and have no loss or visible separation of tooth structure [1]. However, between the lines, there is a hint that we may be facing a more complex issue: “a fracture plane of unknown depth and direction passing through tooth structure that, if not already involving, may progress to communicate with the pulp and/or periodontal ligament” [1]. The reasons for MC formation are known and include abnormalities in the tooth maturation process, excessive or prolonged occlusal forces, traumatic injuries (e.g., during the tooth extraction procedure), temperature variations, dental procedures where a lot of drilling or grinding is required, and removal of brackets at the end of the orthodontic treatment [2,3,4,5,6,7,8]. It has also been shown that the dehydration process can play a critical role in the formation of cracks in the extracted tooth dentin of the root [9,10,11]. Storage of samples in dry conditions creates stress that may lead to significant root damage [11]. However, little is known about whether dehydration would have the same effect on enamel MCs. Still, certain values of enamel MC parameters were found to be unaffected by the dehydration during the preparation of the samples for scanning electron microscopy [8,12].

Enamel MCs can be easily visible to the eye of both the patient and the dentist. In a world where aesthetics plays an important role in people’s daily lives, knowledge of how to look at, react to, and deal with these MCs is even more important. In addition to compromising the aesthetic appearance of the teeth, MCs can damage the integrity of the enamel, lead to the accumulation of stain and plaque on the rough microfractured surface, increase the susceptibility to carious lesions, or cause greater teeth sensitivity after the debonding procedure [3,13,14,15]. The need to document teeth with MCs and to establish monitoring, prevention, and treatment algorithms is becoming evident in everyday clinical practice. The questions of how to improve aesthetics, manage the sensitivity of teeth having MCs, or whether orthodontic treatment should be modified when seeing a patient with multiple teeth with MCs are still unanswered (Figure 1a).

Over the last two decades, the field of MC research has been expanding with advances in laboratory techniques. Methods such as staining and transillumination [16,17,41], which were previously routinely used for MC detection, have been replaced by more sophisticated two-dimensional (2D) techniques that provide information on surface interfaces. The most common methods employed to examine cracks were optical stereomicroscopy (OS), scanning electron microscopy (SEM), and confocal optical profilometry (COP) [2,3,4,5,12,18,19,20,21,22,23,24]. This has made it possible to describe MCs not only qualitatively but also quantitatively. However, in addition to 2D tooth imaging, three-dimensional (3D) scanning methods (e.g., optical coherence tomography (OCT), ultrasound) have increasingly been used to obtain volumetric information of the enamel surface, including cracks [25,27,28]. While the initial interest in tooth MC studies is still growing, there has been an upward trend in the number of publications over the last 7 years towards the use of 3D imaging techniques for tooth structure examination [33,34,35,36,37,38,39,40] (Figure 1b). At the same time, 2D methods are still being employed, but their frequency of use is likely decreasing [3,18,19,20]. The shift from 2D to 3D methods is being driven both by rapidly advancing technologies that allow us to see what could previously only be imagined and by growing scientific interest (Figure 1c,d). New clinical research opportunities are opening up that could meet doctors’ expectations.

X-ray micro-computed tomography (μCT) can be described as a game-changing technology in the field of MC research. In dental studies, this method was introduced in 2007 [42,43], but has only been employed to visualize the inner structure of human tooth enamel in the MC region since 2021 [37]. With previously widely used laboratory techniques for the examination of various crack characteristics (e.g., their number, direction, location, length, and width), it was possible to assess the morphology, position, and dynamics of MCs only on the outer tooth enamel surface [3,4,5,8,19] (Figure 2).

Due to the limitations of the methods employed to analyze the depth of the MCs (e.g., the insufficient penetration and scanning range of the instrument [27,32], sensitivity of the technique to surface curvature [25,32], an indirect method for measuring crack depth [33,40], the necessity to infiltrate contrast material into the cracks in order to examine the depth parameter [40], or the physical evaluation of the crack after the tooth has been sectioned [33]), it remains unclear if MCs are located only within the enamel surface, or if they spread deeper into the tooth, crossing the DEJ and reaching the dentin and even the pulp [38].

The word tomography, derived from the Greek words *tómos* (meaning “slice” or “section”) + *gráphō* (meaning “to write” or “to describe”), hints the working principle in its very name. With a microtomograph, over a thousand 2D photographs rotating a sample are produced in a few hours, from which a high-resolution 3D image of the object (a data cylinder) is reproduced using specialized software. One of the most important features of an X-ray μCT image is the successful separation of structures of different densities in a 3D volume. In 3D, the contrast between objects of different densities is characterized by an excellent resolution of up to one-hundredth of the thickness of a human hair. Modern state-of-the-art laboratory μCTs can achieve resolutions up to 700–1000 nm (4000–10,000 nm is usually the resolution of choice for biological samples) using geometric magnification [37,38,44]. In this way, the components of the tooth can be easily seen: cracks, enamel, dentin, and pulp chamber. This data cylinder can be viewed from different directions and in different sections. It is important to emphasize that at this stage of data processing, artificial intelligence can be successfully employed to understand and automate the spatial arrangement of the structural parts of the tooth. For this purpose, several 2D photographs of the cylinder sections are viewed by eye to determine the location of the objects. Only when the components of the tooth are well identified can the neural network be trained to segment the whole image by means of artificial intelligence and, after correcting the network’s errors and retraining, achieve good segmentation quality over the entire volume of the object [45]. This method thus helps to automate the spatial recognition of the structural parts of the tooth and to understand their position.

Once the samples are properly segmented, it is possible to experiment with custom developed [46,47,48] or commercial [49] 3D visualization software, assuming that each voxel (i.e., the equivalent of a pixel in 2D space) could emit, absorb, or reflect the light. Then, choosing how the voxels should shine or cover each other so that the result looks understandable and attractive when viewed from different directions, it is possible to highlight what is curious, e.g., just the cracks in the enamel of the tooth, the whole network of cracks, or the structures of the dentin surface [37,38].

X-ray μCT marks the non-destructive transition from the outer towards the inner tooth structure, as it allows one to achieve a resolution of a few micrometers and to see the internal structural components of the tooth in much greater detail than in a conventional dental X-ray with the naked eye. Therefore, the application of the X-ray μCT combined with machine learning (ML) for crack analysis has significantly expanded our knowledge of this form of tooth damage [37,38]. This has led to an understanding beyond what was known so far about MCs. Using 3D non-destructive imaging, it has become possible to identify and detect not only enamel MCs that are visible on the outer surface but also those that are buried deep inside the tooth [37,38]. X-ray μCT scanning technique enhanced with convolutional neural network (CNN) assisted voxel classification and volume segmentation (a type of deep learning algorithm) enabled 3D non-destructive examination of all spatially resolved MCs, despite the locations within the tooth in which they originate and extend, as well as the arrangement of MCs and their structural properties [38,50,51,52]. A network of star-shaped MCs (along the main axis of the tooth) inside intact (healthy, undamaged) teeth with visible or absent MCs on the buccal surface has been disclosed, proposing the idea that the cracks might be regarded as on the structural and probably functional components of the tooth [38].

The ability to characterize cracks in 3D raises further question of whether there are changes in the tooth material within the MC compared to the enamel areas that are without cracks, and whether there is a loss of structural integrity at the site of the MC. Knowledge of the extent of damage at the crack area would help to clarify whether and where treatment strategies for teeth with MCs should be targeted. From this perception comes the idea to combine a μCT with photoluminescence (PL) and micro-/nano-Raman spectroscopy (μ-/n-RS) for tooth structural integrity assessment at the MC site (Figure 3).

The optical characteristics of natural teeth [53,54] and the fluorescence differences between sound teeth and teeth affected by caries lesions [55,56] or treated with bleaching agents [57] have already been studied. The photoemission spectra (illuminated with wavelengths of 337.8 nm) of healthy and decayed teeth were found to be different, and the spectral intensity of teeth with dental caries was lower [56]. It was noticed that spectral intensity varied among different carious stages, with deep cavitation displaying the weakest fluorescence [56]. An analysis of the spectral shape derived from the 405 nm laser-induced autofluorescence spectra has also shown it to be a useful tool for diagnosing and staging caries [55]. Although MCs are not classified as tooth decay, it is still unknown whether the PL spectrum at the site of the crack would be similar to that of sound or damaged enamel [58].

So what are the properties of PL and μ-/n-RS that would make these techniques particularly useful for further analysis of the chemical and elemental composition of tooth material? It is important to emphasize that interaction of light with matter shows us the optical properties, composition, and defects of the examined object. This can be achieved using basic non-destructive techniques like PL and μ-/n-RS measurements, which do not require special sample preparation. With PL, the tooth is excited with monochromatic light (laser) and the emission from the radiative process is observed in different areas of the sample. The change in intensity, peak position, and peak full width at half maximum between the good area and the cracked area can be investigated. The spectral analysis yields results on the distribution of hydroxyapatite (HAP) crystals on the outer tooth surface, inside the tooth, and at the depth of light penetration. In addition, the μ-/n-RS method can be used to determine the chemical composition inside each crack and the difference with the good area. Therefore, image measurements are extremely useful tools for analyzing how to change the concentration of chemical bonds inside and outside cracks.

More sophisticated techniques such as atomic force microscopy (AFM) can be employed to examine the surface of teeth samples even at the nano-scale. The topography, roughness, and mechanical properties of the specimen surface can be assessed by means of high-resolution and high-speed imaging [59].

The field of 3D micro-optics and its applications for imaging is rapidly expanding, especially driven by femtosecond laser direct-write 3D multi-photon lithography (MPL, also known as two-photon or multi-photon polymerization) [60]. Some showcasing examples already demonstrate its feasibility for endoscopic use in pre-clinical and clinical conditions [61]. Excellent performance and the small footprint of 3D micro-optics is foreseen to be game changing in enabling diverse sophisticated optical imaging combined with spectroscopic techniques [62] into in situ and in vivo cases for teeth investigation.

Approaching teeth with MCs from individual perspectives and using different but complementary laboratory techniques, we naturally move from 3D to multi-modal tooth imaging (Figure 4).

The volumetric (passive) information of the sample can be supplemented by dynamic (active) image data. It becomes possible to reveal the physical (mechanical) properties of the tooth by scanning under loading conditions or by modifying the environmental parameters (e.g., hydrated via dehydrated medium). Thus, a material responds to the changing surroundings. Some efforts have already been made in this direction, using synchrotron X-ray tomography for the 3D assessment of enamel demineralization in carious lesions [63] and for the analysis of internal enamel during acid exposure [64]. A time-lapse imaging sequence was generated by repeatedly scanning the changes in enamel structure because of acids [64]. Therefore, 3D study of how MCs would behave under compression would give us dynamic information—already in the four-dimensional space.

Apart from X-ray structural analysis, synchrotrons provide high brilliance (brightness) beams at far-infrared (3–30 µm) and THz (30 µm–3 mm) spectral ranges. Spectral brightness has units of time${}^{-1}\;\cdot$ distance ${}^{-2}\;\cdot$ angle ${}^{-2}\;\cdot$ bandwidth ${}^{-1}$ [65]. At those long wavelengths, light penetration into skin depth of ∼λ/4 already yields useful sub-surface characterization in reflection as well as transmission. The use of attenuated total reflection (ATR), especially in the THz spectral range, expands the versatility of THz diagnostics due to the simplicity of sample preparation and measurement, which avoids THz propagation in air where there is a strong THz absorption by water (moisture). Recently, the principle to obtain 2D slicing of optical properties at different sample orientations required for 3D rendering of volume in μCT was demonstrated using ATR [66]. Optical near-field can be used to 3D steer the intensity of far-infrared/THz beams inside the sub-surface of the sample using different polarization azimuth of the incoming beam and analyzing the reflected beam from the sample in the ATR setting [66]. This technique provides access to orientation and anisotropy in absorption or/and birefringence of the tissue, which is present in the teeth [67]. An example of natural birefringent material that also has a linear dichroism at THz wavelengths is bamboo [68]. The four-polarization analysis of transmittance spectra [69] (also applicable to reflectance [70]) provides access to structural anisotropies in absorption and birefringence and also reveals them even when spatial resolution is low [71]. The four-polarization technique is promising for the investigation of structural changes to enamel and dentin caused by ablation with ultra-short laser pulses at a high repetition rate, which is a promising technique for dental repairs and is approaching material removal rates of ∼30 mm${}^{3}$/min and could replace currently used mechanical tools for drilling [72].

Therefore, future perspectives (their strengths, weaknesses, and opportunities) for the analysis of tooth MCs are presented in Figure 5.

It is important to note that destructive tooth examination techniques are being developed in parallel [73]. In order to determine the optimal irradiation conditions for the precise removal of dental hard tissues without structural and compositional damage, the effect of femtosecond laser on the ablation of enamel and dentin was investigated at different pulse wavelengths and pulse intervals [73]. There is a potential to examine not only naturally developed cracks due to mechanical reasons, but also damage to the tooth structure caused by laser processing.

Detailed volumetric imaging of the tooth MCs, which will be complemented by dynamic data in the near future, reveals the need to review and update the definition of crack used until now and to re-assess the role of these MCs and their effect on the structural integrity and longevity of the tooth. It also provides insight into the nature of cracks in natural hard materials and contributes to a better understanding of how bio-inspired structures could be designed to foresee crack propagation in biosolids. Future estimation of the material elemental and chemical composition in a cracked tooth area could lead to the development of new algorithms for the monitoring and treatment of teeth with MCs in daily clinical practice. Considering the quality and quantity of HAP crystals in the MC area, the idea of remineralization therapy for teeth with cracks is foreseen. Finally, artificial intelligence models for segmenting teeth from X-ray μCT images would eliminate the necessity for manual assessment of radiographic data and promote diagnostic efficiency and treatment strategy planning in cracked areas.

Over the past few years, the field of dentistry has witnessed remarkable advancements through the application of various machine learning (ML) models. Initially, studies focused on traditional convolutional neural networks (CNNs) such as U-Net for caries detection and localization [74,75]. Subsequent progress led to the development of more sophisticated architectures, including a 3D-CNN for generating partial dental crowns, demonstrating improved validation accuracy and sensitivity [76]. The evolution continued with the introduction of self-training-based methods, leveraging unlabeled data for student model training, offering computational and performance gains over traditional supervised learning [77]. Hybrid frameworks, combining deep learning (DL) and conventional computer-aided design approaches, demonstrated high accuracy in automatic periodontitis staging [78]. In addition, new artificial intelligence systems have been developed and evaluated to help determine the severity of tooth crowding and the need for tooth extraction when planning orthodontic treatment [79]. The successful combination of DL with orthodontic photographs allowed effective categorization of the tooth crowding problem and good decision-making on the choice of orthodontic treatment strategy [79]. Furthermore, the exploration of variational autoencoders and the application of ML setups for sexual dimorphism showcased a shift towards more complex and versatile models, contributing to enhanced diagnostic capabilities in dental imaging [80,81]. These progressive developments underscore the transformative potential of ML in revolutionizing dental diagnostics and treatment planning.

The utilization of ML techniques in dentistry presents a transformative approach with notable strengths. The implementation of DL models, such as U-Net and 3D-CNN, has exhibited significant precision in tasks like caries detection and partial dental crown generation, showcasing their efficacy in complex dental imaging applications. In recent years, artificial intelligence models have been used mainly for practical applications, in particular, for tooth and alveolar bone segmentation [82] and jaw reconstruction [83,84]. A deep convolutional generative adversarial network (DCGAN) called CTGAN has been proposed to help recover mandibular morphology after disease [83]. A DL approach, based on a variational auto-encoder to extract morphological features, has proven to be a promising tool for reconstructing the shape of the mandible, even when some segments from incomplete images are missing [84]. However, challenges persist, including the dependence on limited and specific datasets, the need for robust external validation across diverse patient populations, and the interpretability of complex models. Overcoming these limitations and addressing ethical considerations will be crucial for the future success and integration of ML in dentistry, promising enhanced diagnostic capabilities and improved patient care. These new research opportunities respond to today’s clinical challenges and are moving rapidly towards the medical field.

## Figures and Tables

**Figure 1 bioengineering-10-01354-f001:**
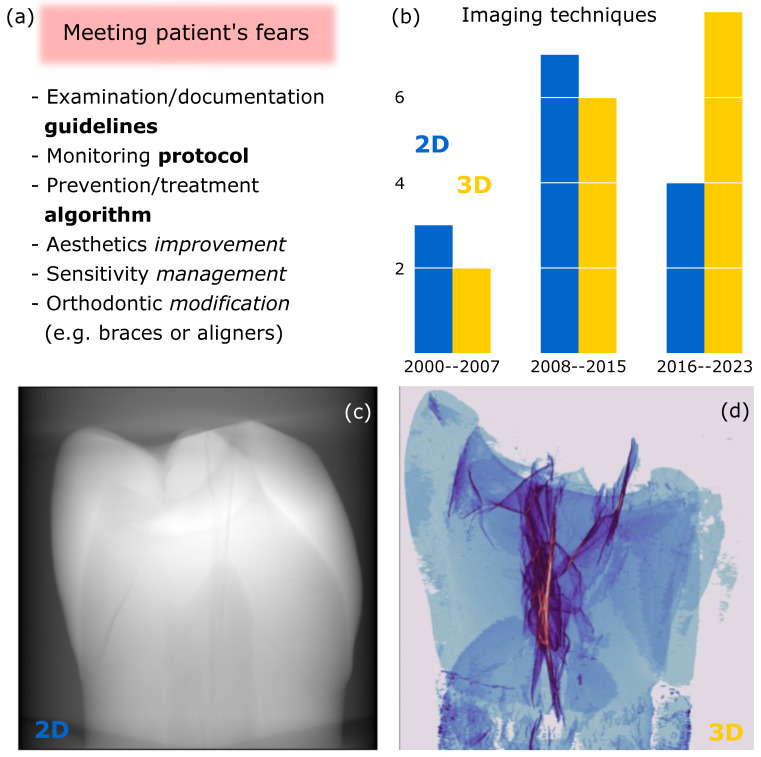
(**a**) Key questions for the clinician who sees a patient with MCs in the enamel; (**b**) overall trends in publications on two-dimensional (2D; blue) [2,3,4,5,12,16,17,18,19,20,21,22,23,24] and three-dimensional (3D; yellow) [25,26,27,28,29,30,31,32,33,34,35,36,37,38,39,40] imaging techniques for teeth MCs from 2000 to 2023; (**c**) X-ray 2D image of tooth; and (**d**) X-ray micro-computed tomography of tooth voxels identified as cracks in 3D and projected in 2D.

**Figure 2 bioengineering-10-01354-f002:**
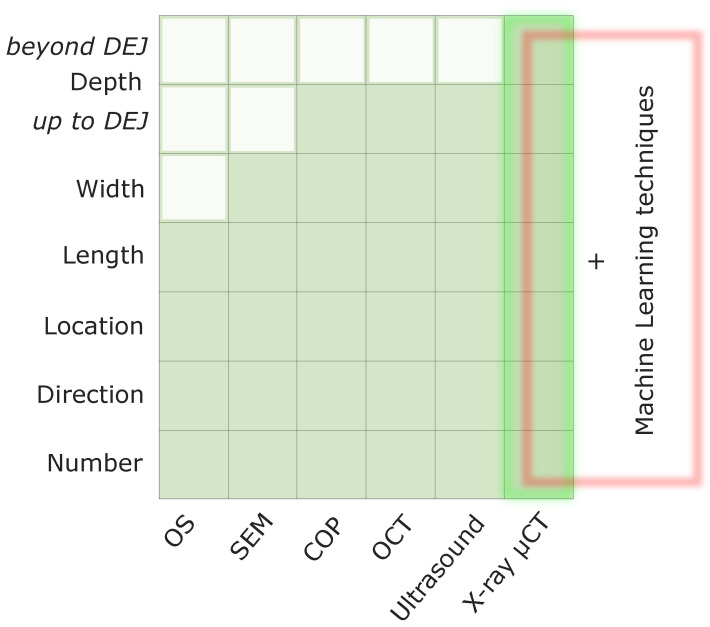
The most common laboratory methods (2D and 3D) for the qualitative and quantitative characteristics of teeth MCs and their ability to measure various cracks parameters. DEJ—dentin–enamel junction, OS—optical stereomicroscope, SEM—scanning electron microscope, COP—confocal optical profilometer, OCT—optical coherence tomography, X-ray μCT—X-ray micro-computed tomography.

**Figure 3 bioengineering-10-01354-f003:**
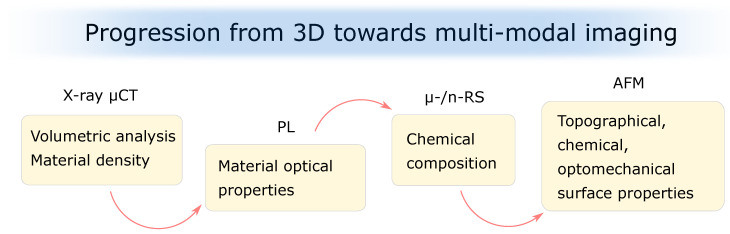
Progression from 3D towards multi-modal imaging of tooth microstructure. Next to the 3D volume definition (X-ray μCT), one can obtain the material’s optical properties (PL), chemical composition (μ-/n-RS), and the topographical, chemical, and optomechanical surface properties (AFM). X-ray μCT—X-ray micro-computed tomography, PL—photoluminescence, μ-/n-RS—micro-/nano-Raman spectroscopy, AFM—atomic force microscopy.

**Figure 4 bioengineering-10-01354-f004:**
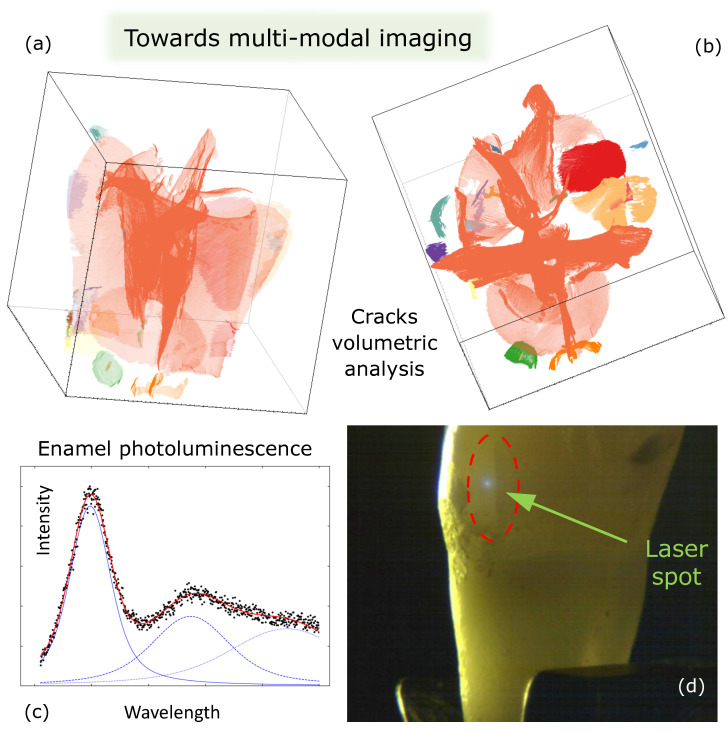
(**a,b**) X-ray micro-computed tomography of tooth voxels identified as cracks in 3D shown in different view perspectives. Each color encodes a unique spatially connected network of cracks. (**c**) Example of photoluminescence (PL) spectra (blue lines show three Gaussian components that are summed to get a red line which represents a best-fit model to the spectra (black dots)) obtained with a laser spot in (**d**).

**Figure 5 bioengineering-10-01354-f005:**
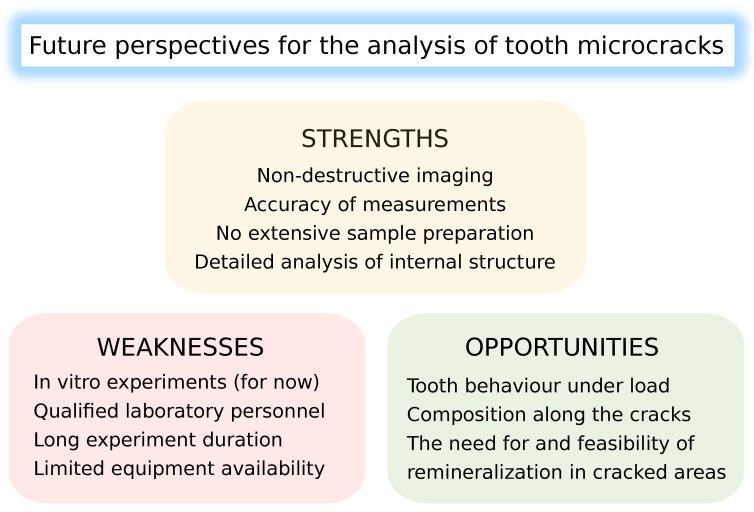
Strengths, weaknesses, and opportunities of novel techniques for tooth MC examination.

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
