# Peer review of "Teeth Microcracks Research: Towards Multi-Modal Imaging"

_bioengineering, 2023, doi:10.3390/bioengineering10121354_

Round 1

Reviewer 1 Report

Comments and Suggestions for Authors

Line #

Comments

4-5

‘all cracks’ – what do we mean by all cracks? Can we qualify with measures and numbers, types?

5

What do we mean regardless of volume of tooth? Did you mean by size or number?

Fig 1(b)

References.  From where the data taken?

92

Pixel in 3D space should be changed to pixel in 2d space

91-92

Need to explain this a bit better. Are you referring to a tool in the 3D visualization software?

Fig 3

Title: 3D imaging of passive coordinates – what do we mean by this?

Add review about

In-situ experiments and experiments performed under some environment

Use of the term 4D

Usually the 4D imaging is referred to time resolved tomography, where you use same imaging technique but vary the independent variables such as time, temperature, enviornment…etc. I suggest that the authors consider changing the name. Perhaps we can say multi-modal investigation?

152-157

There are already some efforts being done in this direction, do the authors aware of these? If so, can they discuss this? Please see references 1-2 below

174-176

I would have liked to see more discussion on the use of artificial intelligence in data analysis and references related to this.

Suggested references

1.      https://www.sciencedirect.com/science/article/pii/S2352492821004104

2.      https://www.mdpi.com/2304-6767/11/5/130

Author Response

Additional corrections:
Reference list has been checked and adjusted.

Again, we are very grateful to Editor and Reviewers for all the comments and suggestions which make the current manuscript version much stronger and more appealing for a general scientific audience. We hope the revised version will be found suitable for publication.

Best Regards,

In the name of all Authors
Irma DumbrytÄ—

Reviewer 2 Report

Comments and Suggestions for Authors

The manuscript entitled
 “Current directions in teeth microcracks research: from 2D via 3D towards 4D imaging”
deals with an important topic, namely microcracks in teeth. The manuscript has its merit, however there are several major concerns.

1.       The term “4D” is generally indicating 3D imaging in time (4th dimension). The better expression would be “multi-modal imaging”.

2.       Synchrotron ?CT imaging provides superior results and should also be mentioned.

3.       It was shown in endodontic research that dehydration is an important factor in crack formation. Address these issues in the manuscript.

Versiani, M.A., Cavalcante, D.M., Belladonna, F.G., Silva, E.J.N.L., Souza, E.M. & De-Deus, G. (2022) A critical analysis of research methods and experimental models to study dentinal microcracks. International Endodontic Journal, 55(Suppl. 1), 178–226. Available from: https://doi.org/10.1111/iej.13660

De-Deus, G, Cavalcante, DM, Belladonna, FG, Carvalhal, J, Souza, EM, Lopes, RT, Versiani, MA, Silva, EJNL, Dummer, PMH. Root dentinal microcracks: a post-extraction experimental phenomenon? International Endodontic Journal, 52, 857–865, 2019.

Title

Many future directions are included in the manuscript, the title should be changed to “Current and future directions…”

Figures 1a and 1b are not needed.

Figure 3 . Add images, obatined with the modalities, listed in the figure legend, namely  X-ray ?CT),optical properties (PL), chemical composition (?-/n-RS), and the topographical, chemical and optomechanical surface properties (AFM). X-ray ?CT - X-ray micro-computed tomography, PL - photoluminescence, ?-/n-RS - micro-/nano-Raman spectroscopy, AFM - atomic force microscopy.

Figure 4 is composed of 4 images. Please mark a,b,c, d and add descriptions and explanations.

References

Refs #41 and #42 are web pages and are not correctly cited.

Comments on the Quality of English Language

NA

Author Response

(The authors gave the same response as above.)

Reviewer 3 Report

Comments and Suggestions for Authors

The authors determined this paper as a perspective article. In my opinion, it is a review and should include more recent articles from the literature and be rewritten as a review.

Author Response

Dear Editor and Reviewers,

Thank you very much for the valuable comments and suggestions.

Amendments were made according to all the mentioned comments. The manuscript was extended and supplemented with additional relevant references. The results and conclusions were presented in more detail. The changes in the manuscript are highlighted in red color. These changes are also highlighted in red in the answers to the Reviewers’ comments.

Please find the detailed list of answers and changes below:

Reviewer 3:
1. The authors determined this paper as a perspective article. In my opinion, it is a review and should include more recent articles from the literature and be rewritten as a review.

Answer to the comment
Thank you for Your comment. Review and perspective articles both provide an overview and analysis of a specific topic or area. However, they differ in their purpose, scope, and tone. The aim of review papers is to summarize and synthesize the existing literature on a given topic, to identify key findings, gaps, and contradictions, and to suggest directions for future research. They often follow a systematic approach to searching, selecting, evaluating, and organizing relevant sources. According to the regulations of publishers such as Springer and Nature, review articles can be narrative (descriptive and qualitative summary of the literature) or quantitative (statistical and numerical summary of the literature). Perspective papers, on the other hand, aim to provide a novel and original approach to a topic, propose a new framework or model, challenge current paradigms or anticipate future trends and implications. Perspective articles are usually focused, subjective, and visionary. They often take a personal and creative approach to express the author's opinion, perspective, or vision.

The field of teeth microcracks research is relatively unexplored. There are comparatively little studies on what more can be done in this area. Only recently have more publications on teeth microcracks appeared and there has been a steady increase in the number of new articles. It is therefore still a growing area. Thus, for the latter reasons, we have classified our work as a perspective paper. However, we would like to stress that we have added new relevant publications to the manuscript in line with the Reviewer's recommendations and improved all the items listed in the "must be improved" section. We certainly agree that it would be useful to write a review article on teeth microcracks in a year or two.

Additional corrections:
Reference list has been checked and adjusted.

Again, we are very grateful to Editor and Reviewers for all the comments and suggestions which make the current manuscript version much stronger and more appealing for a general scientific audience. We hope the revised version will be found suitable for publication.

Best Regards,

In the name of all Authors
Irma DumbrytÄ—

Round 2

Reviewer 1 Report

Comments and Suggestions for Authors

Thank you for the changes. The manuscript looks much better.

Reviewer 3 Report

Comments and Suggestions for Authors

The article can be published in present form.